# Catalytic production of low-carbon footprint sustainable natural gas

Xiaoqin Si [1,3], Rui Lu[1,3], Zhitong Zhao [1], Xiaofeng Yang [1], Feng Wang [1], Huifang Jiang[1,2], Xiaolin Luo[1,2], Aiqin Wang [1], Zhaochi Feng[1], Jie Xu [1] & Fang Lu [1✉]

Natural gas is one of the foremost basic energy sources on earth. Although biological process appears as promising valorization routes to transfer biomass to sustainable methane, the recalcitrance of lignocellulosic biomass is the major limitation for the production of mixing gas to meet the natural gas composition of pipeline transportation. Here we develop a catalytic-drive approach to directly transfer solid biomass to bio-natural gas which can be suitable for the current infrastructure. A catalyst with $Ni_2Al_3$ alloy phase enables nearly complete conversion of various agricultural and forestry residues, the total carbon yield of gas products reaches up to 93% after several hours at relative low-temperature (300 degrees Celsius). And the catalyst shows powerful processing capability for the production of natural gas during thirty cycles. A low-carbon footprint is estimated by a preliminary life cycle assessment, especially for the low hydrogen pressure and non-fossil hydrogen, and technical economic analysis predicts that this process is an economically competitive production process.

[1] State Key Laboratory of Catalysis, Dalian Institute of Chemical Physics, Chinese Academy of Sciences, 116023 Dalian, China. [2] University of Chinese Academy of Sciences, 100049 Beijing, China. [3] These authors contributed equally: Xiaoqin Si, Rui Lu. ✉email: lufang@dicp.ac.cn

Natural gas, one of the global primary energy sources, could be used as fuel for generating electricity, heating, and powering transportation, and also as raw materials to manufacture hydrogen and ammonia[1–4]. $CH_4$ (70–90%) and a small percentage of C2–C4 hydrocarbons constitute primarily of its composition. Biogenic and thermogenic processes are the two widely accepted mechanisms of natural gas genesis[5,6]. No matter generated in which way, several essential conditions for natural gas formation are organic molecules, suitable temperature and atmosphere, a long time of million years and sometimes water is also needed. Abundant and accessible biomass is believed to be the only renewable carbon resource known and a promising alternative to fossil fuels for the production of fuel[7,8] and chemicals[9–12]. Owing to the rapid growth of gas demand and violent fluctuation of price[13,14], utilization of renewable biomass resource to produce synthetic natural gas has attracted great attention.

Anaerobic digestion of plant and animal wastes could produce marsh gas which is primarily composed of $CH_4$ (50–70%) and $CO_2$ (30–50%)[15]. Thermochemical methods have also been developed for the production of biomass fuel gas. The pyrolysis/gasification process, one of the widely used techniques for raw biomass conversion, is usually operated at 600–1000 °C and produces a gas mixture mainly containing CO, $CO_2$, $H_2$, and $CH_4$[16–18]. And the total content of hydrocarbons is generally less than 10%. Coupling the catalytic hydrogenation process with biomass gasification would increase the yield of hydrocarbons[19,20]. The final gas products of the two-step process are composed by 55–75% of $CH_4$ and a fairly higher number of $CO_2$ (15–25%) compared to commercial natural gas. The sub- or supercritical water gasification process is another way for gas production which is usually conducted under the operating conditions of 300–600 °C[21,22]. The content of $CO_2$ in gas products is comparable with or even higher than that of $CH_4$. Overall, the gas produced by all of the above methods could not satisfy the composition requirements of pipeline natural gas. Besides, the remained solid residues after the reaction need to be removed and the separation units would increase energy consumption and operating costs. Therefore, it still remains a challenge for the efficient transformation of biomass to commercial natural gas.

The present study is aimed to produce natural gas by mimicking natural formation conditions. Here, we report a robust catalyst with $Ni_2Al_3$ alloy phase to catalyze solid biomass in aqueous phase to form gas rapidly within just a few hours. And a 93% carbon yield of gas products is achieved at 300 °C after reacting 5 h. The generated gas contains 96% of $CH_4$, 3% of C2–C4 hydrocarbons and only 1% of $CO_2$, which is consistent with the composition of commercial natural gas. The catalyst shows excellent performance for at least 150 h. And the life cycle assessment shows that life cycle primary fossil energy depletion and greenhouse gas emissions in this work can dramatically decrease with the low hydrogen pressure and non-fossil hydrogen.

## Results and discussion

**Catalytic conversion of beech sawdust.** The nickel-based alloy catalyst was prepared through the controlled dealumination process of nickel-aluminum powder and used to convert beech sawdust (detailed in the Methods). In the absence of catalyst, the sawdust remained almost unchanged and existed in the form of solid after the reaction. Compared to commercial nickel catalyst, the prepared nickel-based alloy catalyst showed nearly complete conversion of sawdust and no solid substrate existed in aqueous under the same reaction condition (Supplementary Fig. 1 and Supplementary Table 1). The effect of the initial $H_2$ pressure on the distribution of

gas carbon products was displayed in Supplementary Fig. 2. The carbon yield of gas products increased from 74.0% to 87.2% when the initial $H_2$ pressure increased from 0.1 to 4.0 MPa, and the molar percentage of $CH_4$ in gas products increased from 51.4 to 86.1 mol %, the molar percentage of $CO_2$ decreased from 45.8 to 9.9 mol%. Thus, the content of $CH_4$ in gas products increased with the increasing initial $H_2$ pressure. The reaction results of different temperatures were further depicted in Fig. 1a. As the temperature increased from 250 °C to 300 °C, the total carbon yield was general on the rise from 86.7% to 95.2%. Thereinto, the yield of gas products increased gradually from 59.0% to 87.2%, and the yield of solid products had a slightly decline from 10.4% to 7.9% while the yield of liquid products decreased sharply from 17.3% to 0.1%. Since gas products were the dominant component, the distribution of each certain gas was also illustrated in Fig. 1a and the detailed data were listed in Supplementary Table 1. It was easy to find out that the molar percentage of $CH_4$ in gas products ranged from 86.1 to 91.3 mol%, and the molar percentage of C2–C4 hydrocarbons just fluctuated between 3.7 and 4.5 mol%. That is to say that the composition of gas produced in above method was similar to that of commercial natural gas.

The solid products deposited on the used catalyst were identified by the ultraviolet (UV) Raman spectroscopy analysis and the formed carbon species might be in the form of olefin polymers and benzene derivatives (Fig. 1b). Two-dimensional heteronuclear single-quantum coherence nuclear magnetic resonance (2D HSQC NMR) analysis was used to identify the liquid products which mainly consisted of lignin and polysaccharides[23]. As the reaction temperature increased, the signals of polysaccharides and lignin weakened (Fig. 1c–e). Notably, no signal was shown in 2D HSQC NMR spectrum of reacting at 300 °C. In addition, matrix-assisted laser desorption/ionization time of flight mass spectrometry (MALDI-TOF-MS) analysis showed that the molecular weight of liquid products decreased with increasing the temperature (Supplementary Fig. 3). Above results indicated that the prepared nickel-based alloy catalyst could nearly completely transform solid sawdust to bio-natural gas in high yield, and only few solid carbon species were found as by-products.

**Catalyst characterization and reaction pathway.** X-ray diffraction (XRD) patterns showed that the commercial nickel catalyst only presented the signals of metallic nickel (Fig. 2a). While the prepared nickel-based catalyst revealed the characteristic peaks of $Ni_2Al_3$ alloy phase at 18.1°, 25.3°, 31.2°, 44.6°, and 45.1° except for the signals of metallic nickel. Both of the catalysts had porous structures visualized by scanning transmission electron microscopy (STEM) images (Supplementary Fig. 4). The interplanar spacing of prepared nickel-based catalyst, which is ca. 0.20 nm, was attributed to $Ni_2Al_3$(110) lattice planes (Fig. 2b), while the commercial nickel catalyst revealed Ni(111) lattice planes. Element distribution analysis further illustrated that aluminum was highly dispersed in the alloy catalyst (Fig. 2c). Therefore, it was suggested that the $Ni_2Al_3$ alloy phase existed and was evenly distributed in the prepared catalyst.

In the catalytic conversion of lignocellulosic biomass, it is known that $H^+$ ions generated from the liquid water at elevated temperatures (above 473 K) could catalyse the hydrolysis of biomass constituents to oligosaccharides and lignin[24,25]. Meanwhile, nickel-based alloy catalyst acted in synergy to break the biomass (Supplementary Fig. 5 and Supplementary Table 2), which were further converted to natural gas through the cleavage of C–O and C–C bonds (Supplementary Fig. 6). To explore the unique catalytic behavior of $Ni_2Al_3$ phase toward the reaction pathway over nickel-based alloy catalyst, glucose, xylose, and 4-propylguaiacol, which were the representative units of cellulose,

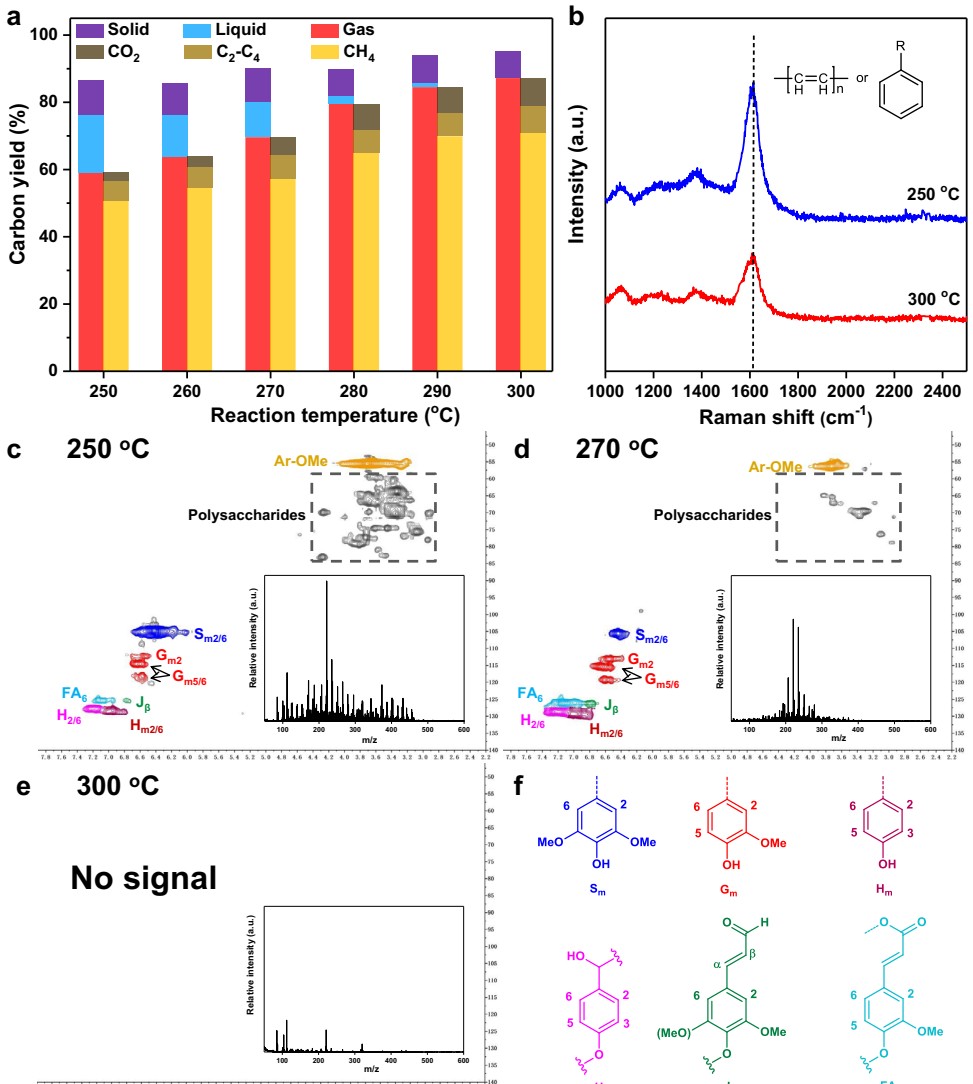

**Fig. 1 Catalytic conversion of beech sawdust over the prepared nickel-based alloy catalyst. a** Carbon yield of solid, liquid, and gas products and the distribution of gas products at various temperatures. **b** Analysis of solid carbon species deposited on the used catalysts by UV Raman spectroscopy. The peak at 1615 cm$^{-1}$ was assigned to the stretching vibration of C=C. **c** 2D HSQC NMR and MALDI-TOF-MS (the inset) analysis of the liquid products after the reaction at 250 °C. **d** 270 °C. **e** 300 °C. **f** The possible lignin structure in the liquid products. Reaction condition: 1.0 g beech sawdust, 5.2 mmol nickel-based alloy catalyst (based on nickel), 20 mL $H_2O$, 4 MPa $H_2$, 5 h. The gas, solid, and liquid products were quantified by gas chromatography, total organic carbon instrument and thermogravimetric analysis, respectively. The carbon yield was calculated according to the mole numbers of carbon in the reaction products and substrates.

hemicellulose, and lignin in biomass, respectively, were converted with nickel-based alloy catalyst. The results exhibited excellent catalytic performance for the production of bio-natural gas (Supplementary Table 3). As abundant hydroxyls were involved in the biomass constituents, methanol and ethanol with hydroxyl groups were chosen as simple model compounds. The gas carbon yield for the conversion of methanol was much higher than that of ethanol, and in contrast to commercial nickel catalyst, the 96.5% and 46.3% yields of methane were obtained from methanol and ethanol conversion with the nickel-based alloy catalyst, respectively (Supplementary Table 4). It demonstrated that the C–O bond was more easily to break than C–C bond in ethanol, and nickel-based catalyst with $Ni_2Al_3$ phase exhibited the high catalytic activity for the cleavage of C–O and C–C bonds. Additionally, the kinetic analysis of methanol and ethanol conversion over the nickel-based catalyst was investigated (Supplementary Fig. 7). The apparent activation energies ($E_a$) of

methanol and ethanol transformation upon using nickel-based alloy catalyst were 114.0 and 121.7 kJ mol$^{-1}$, respectively, which was lower than using commercial nickel catalyst, manifesting the relative catalytic role of $Ni_2Al_3$ alloy in promoting the cleavage of C–O and C–C bonds. Density functional theory (DFT) calculations were performed to study the reaction pathways of C–O and C–C bonds cleavage in ethanol over the nickel-based catalysts (Fig. 2d). It showed that the decomposition of $CH_3CH_2OH^*$ into $CH_3CH_2O^*$ was thermodynamically favored on Ni(111) and $Ni_2Al_3$(110), with energy barriers of 1.04 eV and 0.53 eV, respectively. For the further cleavage of C–O and C–C bonds, two main reaction pathways were evaluated, which were the C–C bond preferentially cleaved ($CH_3CH_2O^* \rightarrow CH_3^* + CH_2O^*$) and C–O bond preferentially cleaved ($CH_3CH_2O^* \rightarrow CH_3CH_2^* + O^*$). However, the process of C-C bond preferentially cleaved ($CH_3CH_2O^* \rightarrow CH_3^* + CH_2O^*$) was thermodynamically hindered on Ni(111) and $Ni_2Al_3$(110) (Supplementary Fig. 8), and

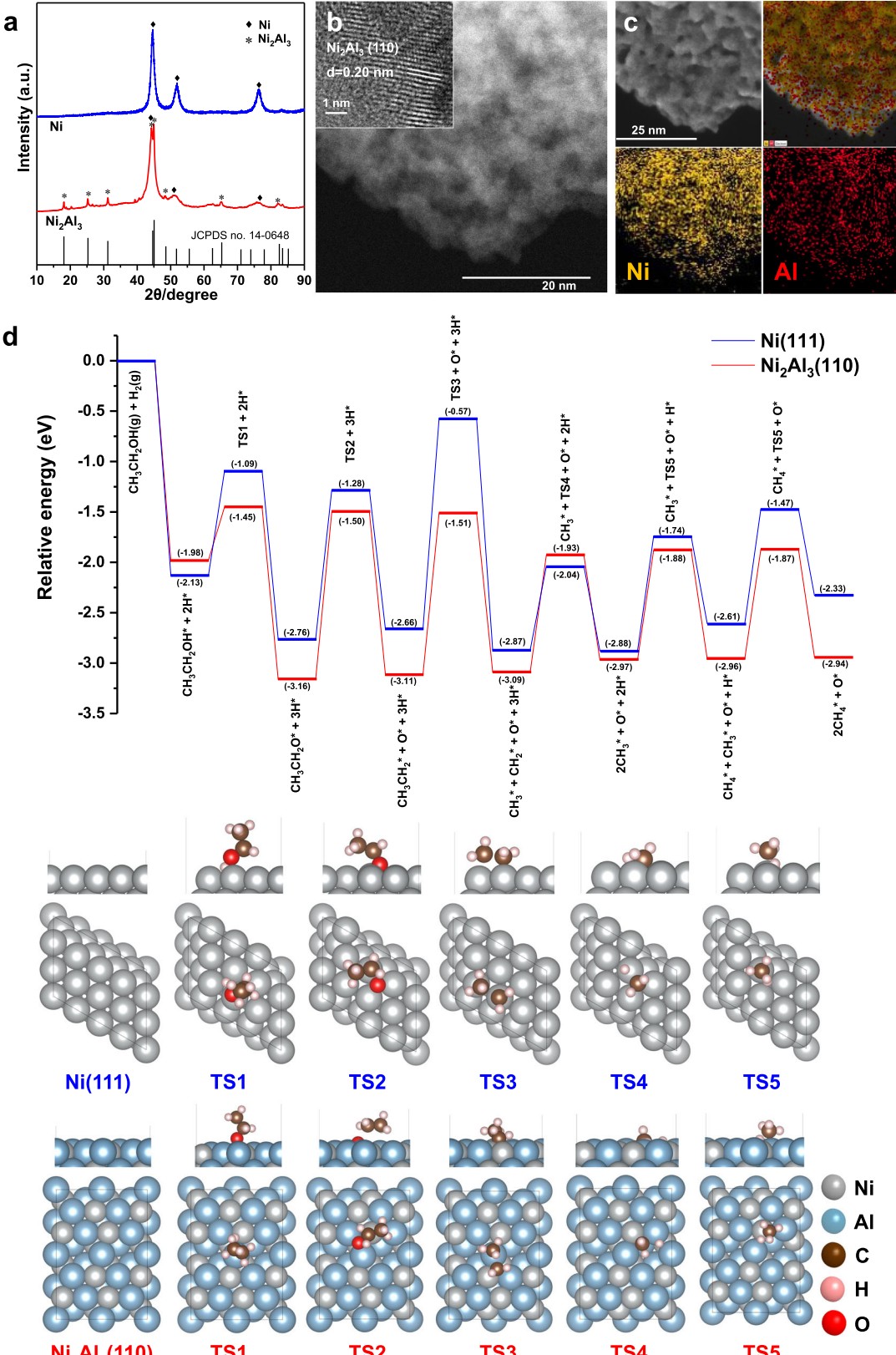

**Fig. 2 Characterization of the nickel-based catalyst with Ni$_2$Al$_3$ alloy phase and reaction pathways of ethanol compound. a** XRD patterns of commercial nickel catalyst and nickel-based catalyst with Ni$_2$Al$_3$ alloy phase (JCPDS card no. 14-0648). **b** High-resolution STEM images of the nickel-based catalyst with Ni$_2$Al$_3$ alloy phase. **c** The energy-dispersive spectrometer (EDS) elemental maps of the nickel-based catalyst with Ni$_2$Al$_3$ alloy phase. **d** Energy profiles for CH$_3$CH$_2$OH dissociation on Ni(111) and Ni$_2$Al$_3$(110) surfaces of C−O bond preferentially cleaved. The $x$ axis shows the reaction intermediates, the $y$ axis shows the relative energy of each state.

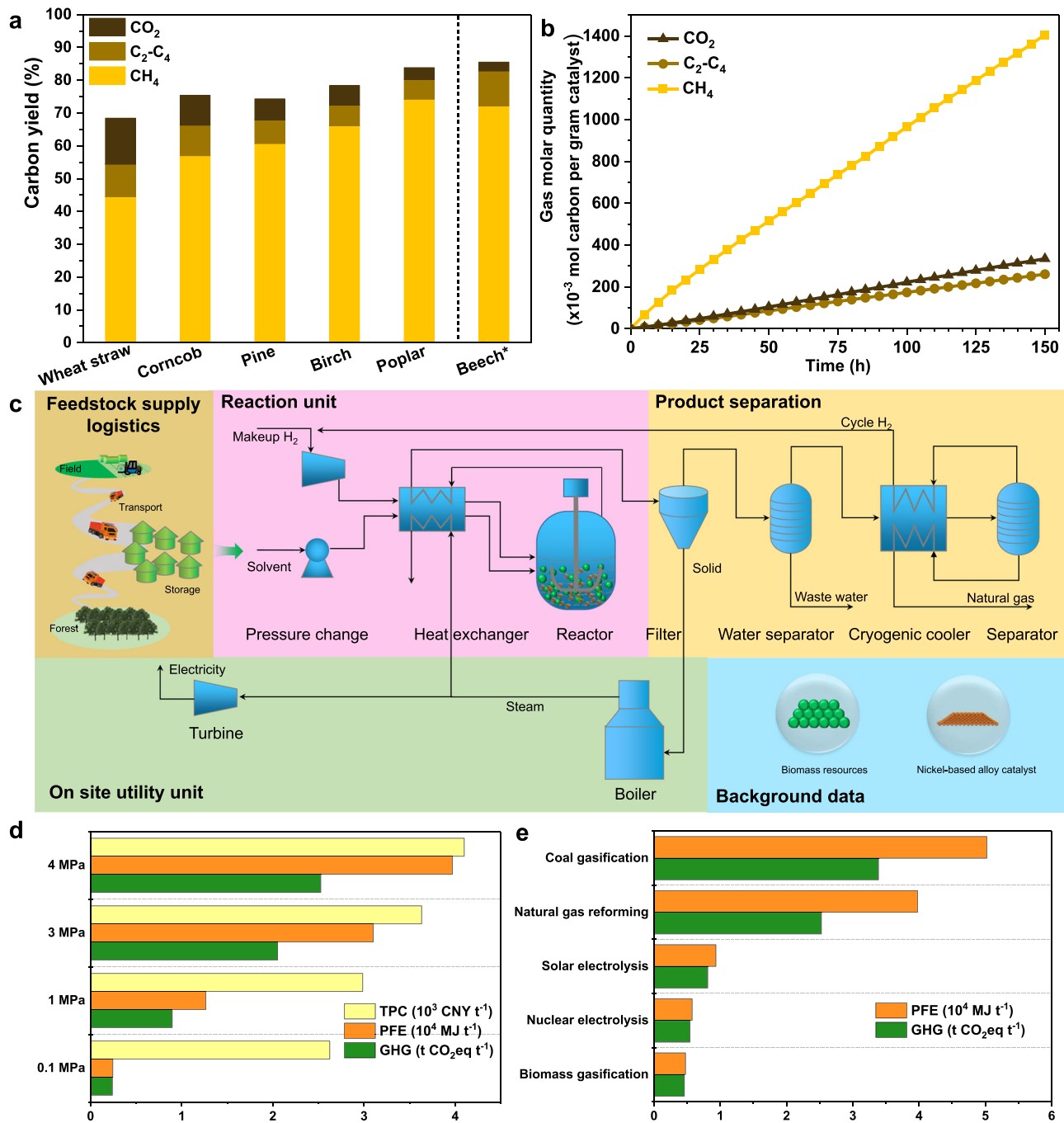

**Fig. 3 Catalytic conversion of various agricultural and forestry residues to bio-natural gas. a** Carbon yield of gas products with various agricultural and forestry residues. **b** Reusability of the catalyst with $Ni_2Al_3$ alloy phase. **c** Conceptual process modeling for bio-natural gas production from biomass. **d** Effects of hydrogen pressure on TEA and LCA of bio-natural gas. **e** Effects of hydrogen source on life cycle PFE depletion and GHG emissions of bio-natural gas. Reaction condition: 1.0 g substrate, 5.2 mmol nickel-based alloy catalyst (based on nickel), 20 mL $H_2O$, 4 MPa $H_2$, 300 °C, 5 h; *10.0 g beech sawdust, 52.0 mmol nickel-based alloy catalyst (based on nickel), 200 mL $H_2O$.

the energy barriers reached up to 2.14 eV and 2.41 eV, respectively. In contrast, the process of C–O bond preferentially cleaved ($CH_3CH_2O^* \rightarrow CH_3CH_2^* + O^*$) was more feasible (Fig. 2d). Furthermore, the energy barrier for $CH_3CH_2^* \rightarrow CH_3^* + CH_2^*$ process on $Ni_2Al_3(110)$ was 1.60 eV, which was much lower than that of 2.09 eV on Ni(111). Thus, the above results demonstrated that the preferential cleavage of C–O bond in $CH_3CH_2O^*$ on aluminum sites of $Ni_2Al_3(110)$ might promote the further cleavage of C–C bond in $CH_3CH_2^*$ intermediate.

**Conversion of agricultural and forestry residues**. The agricultural and forestry residues including straw, corncob, and various sawdusts were also used in the developed process (Fig. 3a, Supplementary Fig. 9, and Supplementary Table 5). Wheat straw and corncob gave 68.4% and 75.4% yield of gas products, respectively. A gas yield of 74.3% accompanied with the $CH_4$ molar percentage of 86.7 mol% was obtained by the conversion of pine sawdust. When using hardwood sawdust such as birch and poplar, the gas yield could be achieved 78.4% and 83.8%, respectively. According to the results of each component

conversion in biomass, the polysaccharides including cellulose and hemicellulose were more easily converted to bio-natural gas than that of lignin (Supplementary Table 6). Therefore, the higher yield of gas products in the conversion of forestry residues may be ascribed to the richer content of cellulose and hemicellulose[26]. A concise Sankey diagram was outlined based on carbon balance of the optimized reaction result with beech sawdust. It showed that 93% of organic carbon in beech sawdust could transform into gas products in which the yield of $CH_4$, C2–C4 hydrocarbons and $CO_2$ reached to 85%, 7%, and 1%, respectively (Supplementary Fig. 10). On account of this proposed process, we predicted that 1 kg of raw biomass could produce 780 L bio-natural gas which contained 96% of $CH_4$, 3% of C2–C4 hydrocarbons and only 1% of $CO_2$. The nickel-based catalyst with $Ni_2Al_3$ alloy phase was tested for its natural gas production capability, and the total carbon mole numbers of $CH_4$ and C2–C4 hydrocarbons steadily increased during 30 cycles for 150 h (Fig. 3b). Tenfold of scaling up the experiment with beech sawdust as substrate could acquire gas yield of 85.5% and $CH_4$ molar percentage of 91.2 mol%, similar to the results of small-sized experiment (Fig. 3a), which was permitted to propose a potential industrial-scale process for the catalytic conversion of lignocellulosic biomass to natural gas.

On the basis of our experimental data, the conceptual design and process modeling, including reaction, product separation, on-site utilities, and heat integration, were performed (Fig. 3c). Subsequently, the techno-economic analysis (TEA) and life cycle assessment (LCA) were conducted to evaluate economic and environmental potential. The production cost of natural gas in the current work was ~4100 CNY/t (Supplementary Table 9), which was in the range of fossil-derived natural gas prices between 2500 and 4500 CNY/t over the last year in China[27]. The life cycle primary fossil energy (PFE) depletion and greenhouse gas (GHG) emissions could be reduced by 26.3% and 34.1% compared to the fossil-natural gas (Supplementary Fig. 13). The expense of hydrogen accounts for the predominant contribution to the production cost of bio-natural gas (Supplementary Figs. 14 and 15). According to the element composition of beech sawdust and our experimental results, the optimized amount of hydrogen consumption was 1.6 per generated carbon in gas products (Supplementary Table 10), which was less than that of industrial coal-to-natural gas process[28]. The effect of $H_2$ pressure on TEA and LCA of bio-natural gas was further investigated (Fig. 3d). As $H_2$ pressure in this catalytic reaction decreased from 4.0 to 0.1 MPa, the production cost of natural gas decreased from 4100 to 2625 CNY t$^{-1}$, meanwhile, the PFE depletion and GHG emissions were generally reduced to only 6.2% and 9.5% of that with 4 MPa hydrogen for bio-natural gas. The results confirmed a sensible decrease of the environmental burdens exploiting renewable biomass for the sustainable process, and promoted the economic development for the low carbon future.

Sustainable production of hydrogen, for example, water electrolysis[29], has experienced a distinguished development in recent years and the manufacturing cost derived from abandoned light and wind has a sharp decrease[30]. We further investigated the effect of hydrogen source, such as coal gasification, natural gas reforming, solar electrolysis, nuclear electrolysis and biomass gasification, on the life cycle PFE depletion and GHG emissions of bio-natural gas (Fig. 3e). It revealed that the production of bio-natural gas using $H_2$, which was derived from solar electrolysis, nuclear electrolysis and biomass gasification, could reduce 81.4–90.5% of PFE depletion and 76.1–86.5% of GHG emissions compared to coal gasification-based $H_2$. Thus, the production of bio-natural gas, which comes from the combined non-fossil hydrogen with lignocellulosic biomass, can be further applied for industry, commercial building, residential house, transportation, and electric power plant by the existing transportation pipelines (Fig. 4).

In comparison with the reported literatures for the production of natural gas, our process has the following features:

(1) The process could accomplish nearly complete conversion of various agricultural and forestry residues to bio-natural gas over the accessible and low-cost catalyst with relatively low operating temperature. Moreover, the composition of gas products is basically consistent with commercial natural gas.

(2) The technical economic analysis predicts that this process is an economically competitive production process, and a low-carbon footprint of this effective technology is estimated by a preliminary life cycle assessment.

We develop an efficient technology to enable nearly complete conversion of raw biomass to bio-natural gas with carbon molar yield reaching up to 93%. And the generated gas contains 96% of $CH_4$, 3% of C2–C4 hydrocarbons, and only 1% of $CO_2$, which is suitable for the current pipeline transportation of natural gas. The wonderful bio-natural gas production performance may be attributed to the unique ability of $Ni_2Al_3$ phase to induce the preferential dissociative adsorption of hydroxyl groups on aluminum sites, which probably facilitates the cleavage of C–O bonds and further C–C bonds in lignocellulose. Meanwhile, the catalyst shows powerful processing capability for the production of natural gas in recycling. And the life cycle assessment shows that life cycle primary fossil energy depletion and greenhouse gas emissions can dramatically decrease with the low hydrogen pressure and non-fossil hydrogen. Combined with further process development, this efficient approach may therefore play an important role in reducing global carbon dioxide emissions as well as in the formation of bio-natural gas.

## Methods

**Chemicals and materials**. Wheat straw and corncob were obtained from Anhui Province, China. Pine, birch and poplar were obtained from Liaoning Province, China. And beech was provided by Dansk Træmeland. The all feedstocks were dried at 110 °C for 24 h and milled into powder with the size <40 meshes before use. Commercial nickel and nickel-aluminum powders were purchased from Dalian Tongyong Chemical Co., Ltd. Methanol (>99%) was purchased from TCI. Ethanol (>99%) and glucose (AR) were purchased from Tianjin Kemiou Chemical Reagent Co., Ltd. DMSO-$d_6$ (>99.9% D) and 4-propylguaiacol were purchased from Sigma-Aldrich. Microcrystalline cellulose and xylose were purchased from Alfa Aesar. Xylan was purchased from Shanghai Yuanye biotechnology Co., Ltd. Methane (>99%), ethane (>99%), propane (>99%), butane (>99%), hydrogen (>99%), carbon dioxide (>99%), and the mixed standard gas were provided by Dalian GuangMing Special Gas Products Co., Ltd. Milli-Q water was used in all the experiments.

**Preparation of the nickel-based alloy catalyst**. The nickel-based alloy catalyst was prepared by activating nickel-aluminum powder through the controlled dealumination process. The detailed procedure was as follows: 2.0 g of nickel-aluminum powder was introduced into the round-bottom flask. 30 g of 20 wt% NaOH aqueous solution was added dropwise to the reactor and the mixture was stirred at 90 °C for 1 h. Then the reaction mixture was cooled down to room temperature, the solid powder was centrifuged and washed to neutral with water, and stored in water before use.

**Catalytic conversion of raw biomass**. The catalytic conversion of raw biomass was carried out in a 50 mL stainless steel autoclave (T316 Stainless Steel, Parr Instrument), the biomass substrates contained methanol, ethanol, glucose, xylose, 4-propylguaiacol, microcrystalline cellulose, xylan, organosolv lignin, wheat straw, corncob, pine, birch, poplar, and beech sawdust. Typically, 0.2–1.0 g of biomass substrates, 0.15–0.40 g of catalysts, and 20 mL of water were added into the reactor. The reactor was purged with $N_2$ for three times to expel air and sealed, further purged with $H_2$ for five times to expel $N_2$, and filled with 4.0 MPa $H_2$ at room temperature. Then the autoclave was heated to 250–300 °C within 35–55 min and maintained for 2–5 h with a stirring speed of 800 rpm. After the reaction, the autoclave was rapidly cooled down to 150 °C in air and further cooled down to the room temperature in the ice-water bath. And the monitored pressure and temperature were recorded. The gas products were collected with the sampling bag for further analysis, the catalyst was separated by the magnetic bar, and the liquid products were stored for further analysis. The rates of reaction for methanol or ethanol were calculated based on the rate of methane production in a certain reaction time, whereas the other side reactions were neglected. The apparent activation energy was evaluated from the slope of Arrhenius plots between the ln of reaction rate against 1000/T.

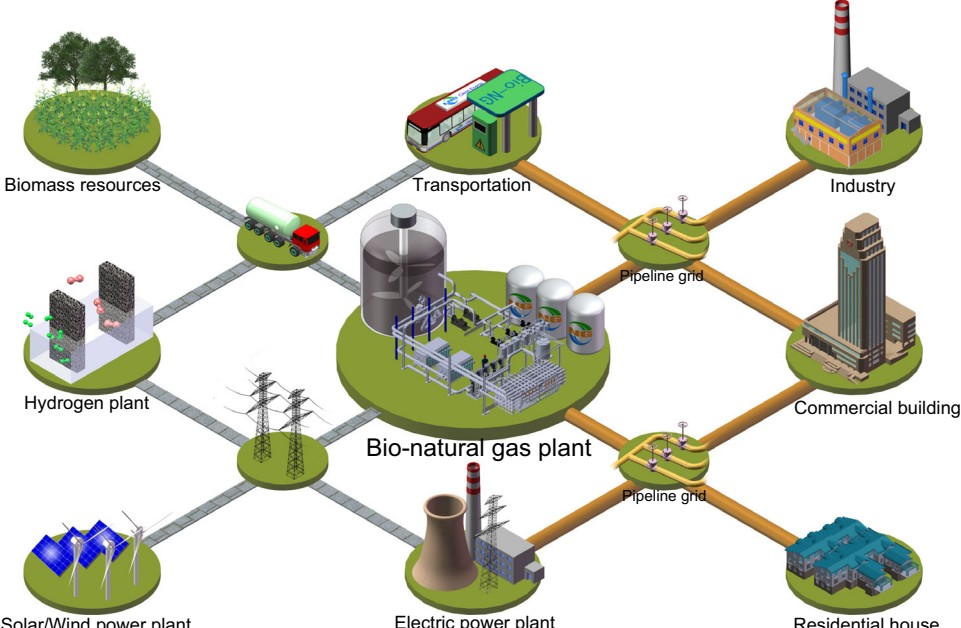

**Fig. 4 Conceptual diagram for bio-natural gas production from biomass.** It is comprised of the production and application of bio-natural gas.

For the reusability of the prepared nickel-based alloy catalyst, the gas products were collected with the sampling bag after the reaction. For the next cycle, the catalyst was separated by a magnetic bar, and then 1.0 g of beech sawdust, 20 mL of water and 0.38 g separated catalyst containing a trace added catalyst for the loss was introduced into the reactor. Then the reactor was sealed and purged with $N_2$ three times to expel air, further purged with $H_2$ five times to expel $N_2$, and filled with 4.0 MPa $H_2$ at room temperature. The autoclave was heated to 300 °C within 55 min and maintained for 5 h with stirring at 800 rpm. After the reaction, the autoclave was cooled down to room temperature, and the gas products were collected with the sampling bag for further analysis again.

**Analysis of gas and liquid products**. Gas products were analysed off-line by using tandem gas chromatography (GC) with thermal conductivity detector (TCD), flame-ionization detector (FID), and standard gas mixtures as external standards. Hydrogen and methane analysis were performed on an Agilent 7890A GC system using an Agilent 19095P-MS0 capillary column. And hydrogen was quantified on the Agilent 7890A GC system. The column was kept at 80 °C for 14 min. The inlet temperature was set at 250 °C. The carrier gas was Ar with a flow rate of 30 mL min$^{-1}$. Methane, ethane, propane, butane, and carbon dioxide analysis were performed on an Agilent 7890B GC system using a HayeSep Q capillary column and a MolSieve 5A with TCD and FID detectors. The column was initially held at 80 °C for 5 min, then ramped to 160 °C at a rate of 20 °C min$^{-1}$, and kept at 160 °C for 5 min. The inlet temperature was set at 250 °C. The carrier gas was He with a flow rate of 30 mL min$^{-1}$. Methane, ethane, propane, and butane were quantified on the Agilent 7890B GC system with FID detector. And carbon dioxide was quantified on the Agilent 7890B GC system with TCD detector. The results obtained from GC analysis were the volume percentage of mixed gas and further transformed into the mole numbers of each component using the Ideal Gas Equation.

The organic carbon of liquid products was measured by DR 3900 Organic Carbon Instrument. The reaction liquid was diluted with water, and then the concentration of organic carbon was measured to obtain the total organic carbon (TOC) in the liquid products.

The carbon molar yield was calculated based on the carbon mole numbers in substrates and products, which was defined as follows:

$$\text{Carbon molar yield} = \left( \frac{\text{Carbon mole numbers of products}}{\text{Carbon mole numbers of lignocellulose substrate}} \right) \times 100\% \quad (1)$$

The gas products distribution was calculated according to the mole numbers of each gas and the total mole numbers of gas products, which was defined as follows:

$$\text{Gas products distribution} = \left( \frac{\text{Mole numbers of each gas}}{\text{Total mole numbers of gas products}} \right) \times 100\% \quad (2)$$

**Elemental analysis**. The content of C and S in lignocellulose substrates was measured on a Horiba EMIA-8100. The content of H, O, and N in lignocellulose substrates was measured on a Horiba EMGA-930.

**XRD analysis**. Powder X-ray diffraction (XRD) was performed on a Rigaku D/Max 2500/PC diffractometer with Cu Kα radiation ($\lambda = 0.15418$ nm) operated at 40 kV/200 mA. And the wide-angle patterns were recorded from 10° to 90° (2θ) at a scan rate of 2.5° min$^{-1}$.

**STEM characterization**. High-resolution scanning transmission electron microscopy (STEM) and EDS elemental maps were performed on a Hitachi HF 5000 STEM/TEM operated at 200 kV.

**UV Raman spectroscopy**. Ultraviolet (UV) Raman spectroscopy was performed on a home-assembled UV Raman spectrograph equipped with a Jobin-Yvon T64000 triple-stage spectrograph. The double frequency of Coherent Verdi-V10 laser through WaveTrain CW frequency doubler was used as the excitation source for the laser line at 244 nm.

**2D HSQC NMR spectroscopy**. Two-dimensional heteronuclear single-quantum coherence nuclear magnetic resonance (2D HSQC NMR) spectroscopy was carried out on a Bruker ADVANCE III 400 MHz spectrometer at 25 °C. After the reaction, a 6.5 mL of the reaction liquid was added into the flask, and water as solvent was removed by rotary evaporator. The left products were mixed with 0.5 mL of DMSO-$d_6$. The HSQC spectra were processed with MestReNova software, and the peak of DMSO solvent was used as an internal reference point (δC/δH 39.52/2.49). The cross-peaks of 2D HSQC NMR spectra were assigned following the related publications[23,31].

**Thermogravimetric analysis**. Thermogravimetric analysis was performed on a NETZSCH STA 409PC apparatus to measure the carbon deposition on the solid catalysts, which was conducted under air flow from 100 °C to 800 °C at a rate of 10 °C min$^{-1}$. Before the test, the solid catalysts were separated by magnetic attraction and dried at 110 °C for 24 h in the vacuum oven.

**MALDI-TOF-MS analysis**. Matrix-assisted laser desorption/ionization time of flight mass spectrometry (MALDI-TOF-MS) analysis was carried out on an AB SCIEX TOF/TOF 5800 instrument (AB SCIEX, Shanghai, China) using a neodymium: yttrium aluminum garnet (Nd: YAG) laser with the 355 nm wavelength in positive reflective modes. A 1 μL of the reaction liquid was pipetted onto a stainless-steel plate and dried under ambient conditions, then, a 0.5 μL of 2,5-dihydroxybenzoic acid (DHB) matrix suspension was dropped onto the layer of reaction liquid and further dried under ambient conditions. Liquid products were analyzed by MALDI-TOF-MS to detect the change of molecular weight after the reaction.

**DFT calculations**. The Vienna Ab Initio Package (VASP)[32] was employed to perform all the spin-polarized DFT calculations within the generalized gradient approximation (GGA) in the PBE[33] formulation. The projected augmented wave (PAW) potentials[34,35] was chosen to describe the ionic cores and take valence electrons into account using a plane-wave basis set with a kinetic energy cutoff of

400 eV. Partial occupancies of the Kohn–Sham orbitals were allowed using the Methfessel-Paxton smearing method and a width of 0.20 eV. The electronic energy was considered self-consistent when the energy change was smaller than $10^{-5}$ eV. Geometry optimization was considered convergent when the energy change was smaller than $10^{-6}$ eV. Grimme's DFT-D3 methodology[36] was used to describe the dispersion interactions.

The equilibrium lattice constants of hexagonal $Ni_2Al_3$ unit cell were optimized, when using a $11 \times 11 \times 9$ Monkhorst-Pack k-point grid for Brillouin zone sampling, to be $a = b = 4.000$ Å, $c = 4.859$ Å, $\alpha = \beta = 90°$, and $\gamma = 120°$. Then it was used to construct a $Ni_2Al_3(110)$ surface model with $p(2 \times 2)$ periodicity in the $x$ and $y$ directions and 4 atomic layers in the $z$ direction by vacuum depth of 15 Å in order to separate the surface slab from its periodic duplicates. This $Ni_2Al_3(110)$ surface models are comprised of 32 Ni and 48 Al atoms. During structural optimizations, a $2 \times 3 \times 1$ k-point grid in the Brillouin zone was used for k-point sampling, and the top two atomic layers were allowed to fully relax while the bottom two were fixed.

The equilibrium lattice constant of FCC Ni unit cell was optimized, when using a $15 \times 15 \times 15$ Monkhorst-Pack k-point grid for Brillouin zone sampling, to be $a = 3.476$ Å. Then it was used to construct a Ni(111) surface model with $p(4 \times 3)$ periodicity in the $x$ and $y$ directions and 4 atomic layers in the $z$ direction by vacuum depth of 15 Å in order to separate the surface slab from its periodic duplicates. This Ni(111) surface models comprises of 48 Ni atoms. During structural optimizations, a $2 \times 3 \times 1$ k-point grid in the Brillouin zone was used for k-point sampling, and the top two atomic layers were allowed to fully relax while the bottom two were fixed.

The adsorption energy ($E_{ads}$) of adsorbate A was defined as follows:

$$E_{ads} = E_{A/Surf} - E_{Surf} - E_{A(g)} \quad (3)$$

where $E_{A/Surf}$, $E_{Surf}$, and $E_{A(g)}$ are the energy of adsorbate A adsorbed on the surface, the energy of clean surface, and the energy of isolated A molecule in a cubic periodic box with a side length of 20 Å and a $1 \times 1 \times 1$ Monkhorst–Pack k-point grid for Brillouin zone sampling, respectively.

Finally, transition states for elementary reaction steps were determined by a combination of the nudged elastic band (NEB) method[37] and the dimer method[38–40]. In the NEB method, the path between the reactant and product is discretized into a series of structural images. The image that is closest to a likely transition state structure was then employed as an initial guess structure for the dimer method.

**Process design details**. The process model was designed based on the experimental data, which is divided into three major sections: reaction, product separation and on-site utility units. Wood serves as feedstock with an annual consumption of 500 kilo metric ton. The wood and water are mixed with certain ratio paralleled with experimental data, and then heated at 300 °C in the heat exchanger. The feedstock was fed into the reactor where 12.4 MPa $H_2$ was filled (Supplementary Fig. 11). The preheated mixture was then converted into methane, ethane, propane and butane in the reactor, and the remainder was assumed as char. The crude product flow was quenched in the heat exchanger and then entered into a char remover. Herein, char was separated from the crude product. The char remover was followed by a water separator where the temperature was set at 38 °C and 99.99% of water was separated from product flow. After then, the components of the product flow include $H_2$ (33.6 mol%), methane (63.4 mol%) and other hydrocarbon gas. To improve $H_2$ efficiency, a cryogenic separation process was designed to recycle $H_2$ with the purity of 99%. The remainder gas mixture was natural gas. Note that char as low-value product, was delivered to boiler to generate steam. Steam was supplied for the whole plant, and excess steam (if any) was used to generate electricity.

The aspen plus was used to simulate the whole process designed to obtain the steady-state material and energy flows. Specifically, feedstock wood, char and ash were defined as nonconventional components. Proximate and ultimate analysis of wood is shown in Supplementary Table 7. HCOALGEN and DCOALIGHT methods were used to calculate the enthalpy and densities of nonconventional components. The reactor was simulated with Ryield block and product distribution was paralleled with experimental data. The char remover and water separator were simulated with Sep and Flash2 blocks, and the cryogenic separation process consist of Flash2 and HeatX block. The Peng-ROB method was selected as the property method. The major flow results are shown in Supplementary Table 8.

**Life cycle assessment**. LCA is used to evaluate PFE depletion and GHG emissions of bio-natural gas. GHGs include $CO_2$, $CH_4$, and $N_2O$ and quantified as a 100 year-time horizon with the units of kg $CO_2$ equivalent[41]. The PFE consists of primary energy coal, petroleum and natural gas, which is measured based on corresponding lower heating value (LHV), as shown in Eqs. (4 and 5).

$$GHG = E_{CO2} + 25E_{CH4} + 298E_{N2O} \quad (4)$$

$$PFE = (LHV_iC_i) \quad (5)$$

where $E_{CO2}$, $E_{CH4}$, and $E_{N2O}$ represent emissions of $CO_2$, $CH_4$, and $N_2O$, respectively, kg; $LHV_i$ is the lower heating value of primary fossil energy $i$, MJ/kg; $C_i$ is consumption amount of primary fossil energy $i$, kg; Subscript $i$ represent primary fossil energy, and can be coal, petroleum, and natural gas.

The functional unit is 1 metric ton of natural gas. The system boundary covers conversion process, combustion process, and all upstream of feedstock required materials and utilities (Supplementary Fig. 12). Specifically, life cycle system boundary of wood-derived natural gas includes wood collection, hydrogen production, electricity generation, conversion, and combustion process, in which the wood collection process involves tree farming, fertilizer application, harvest, and transport. Note that hydrogen production was a hotspot and many sources has been reported, such as coal gasification, natural gas reforming, biomass gasification, biomass reforming, biomass electrolysis, nuclear thermochemical, nuclear electrolysis, solar electrolysis, and wind electrolysis. Natural gas reforming has been used in view of its widespread application at present, which can identify opportunities and constraints associated with the current work other than advanced hydrogen production pathway. Electricity generation is paralleled with grids in China, including 70.1% of coal-fired power, 19.3% of hydroelectric power, 2.5% of natural gas-fired power, 2.9% of nuclear power, and 5.2% of other power.

Greet 2018 software is used to evaluate the LCA of bio-natural gas[42]. Inventory of conversion of wood to natural gas was shown in Supplementary Table 8 as mentioned above. Tree farming, fertilizer application, harvest and transport, hydrogen production, and electricity generation were obtained from inherent database of Greet software.

In addition, LCA of fossil-natural gas was used as a reference to compare the environmental benefits of the bio-natural gas. The production inventory of natural gas was taken from an inherent database of Greet software and combustion factors are taken from the report of Intergovernmental Panel on Climate Change[41].

**Techno-economic analysis**. TEA was evaluated with total capital investment (TCI) and total production cost (TPC). The total capital investment was estimated by installed costs of all equipment. Herein, the installed cost of the cryogenic system was calculated by the "six-tenths rule" based on previous works, as Eq. (6)[43]. While the installed costs of other equipment was estimated based on the calculated flowrates and the experimental information, such as acidity and the employed residence times with the Aspen Process Economic Analyzer. All capital investment of the equipment was adjusted to 2018 CNY with location factor of 0.61[44] and Chemical Engineering Plant Cost Index (CEPCI)[45].

$$\frac{C_2}{C_1} = \left(\frac{S_2}{S_1}\right)^{0.6} \quad (6)$$

where $S$ represents process scale, $C$ represents capital cost, subscript 1, 2 represent different plants, 0.6 represents scale factor.

The total production cost includes raw materials, utilities, operating and maintenance, depreciation, plant overhead, administrative, and distribution and selling costs[46]. The detail parameters and assumptions for the estimation of total capital investment and total product cost are depicted in Supplementary Table 9.

## Data availability

The authors declare that the data supporting the findings of this study are available in the article and Supplementary Information. Additional datasets related to this study are available from the corresponding authors on reasonable request.

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

## Acknowledgements

We received financial support from the National Key R&D Program of China (grant 2018YFB1501600 to F.L.), the National Natural Science Foundation of China (grants 21872139 to F.L. and 21908218 to R.L.), the Strategic Priority Research Program of the Chinese Academy of Sciences (grant XDB17020300 to F.L.), and DICP Grant (grant I201944 to F.L.).

## Author contributions

X.S. designed the research and carried out most of the experiments, and wrote the manuscript. R.L. discussed and revised the manuscript. Z.Z. carried out the life-cycle assessment. X.Y. conducted the DFT calculation. F.W., A.W., and J.X. gave useful advices to the experiment. H.J. and X.L. contributed to product analysis. Z.F. contributed to UV Raman spectroscopy analysis. F.L. conceived the research.

## Competing interests

The authors declare no competing interests.
