## [Peer Review File · Nature Communications]

Title: Catalytic Production of Low-carbon Footprint Sustainable Natural GasREVIEWER COMMENTS

Reviewer #1 (Remarks to the Author):

In this paper, authors claim to have come up with a novel catalytic method to convert lignocellulosic biomass into natural gas. Though the topic is of great interest and some of the data presented by authors looks promising, I cannot recommend this article for publication in its present form for the following reasons:

1) Any form of lignocellulosic Biomass contains about 30-40 wt% of oxygen and when you try to transform that biomass into any form, that oxygen is going to be present in one of the products/by-products (to complete oxygen balance). However, I was intrigued to see the data in Fig. 1, where after the process, at 300, authors claim that they have about 90% yield of gas that contains mostly C1-C4 hydrocarbons and little amount of CO₂. I fail to understand where does the large amount of oxygen that is present in the biomass go??? Based on the yield of solid products and some characterization they have shown, its not possible to have all that oxygen in the solid. This raises serious concern about their analysis methods. Authors must demonstrate clearly where is the oxygen gone.

2) Though the process is interesting and intriguing, there is little science presented in the MS. For example, the DFT calculations are absolutely of no use. In the experiment, they show that without catalyst, the sawdust remains in the liquid solution and with catalyst, you see it getting converted. And then they show DFT data showing some elementary steps of ethanol dissociation (without any kinetics though) and claim that those steps are favored thermodynamically. However, catalyst's job is not to alter the thermodynamics but to alter the kinetics. Yes, some steps could be downhill in energy on one catalysts vs the other, but overall thermodynamics cannot be altered by the catalyst. Authors need to clearly show how kinetics of the steps involved in biomass molecules transformation into smaller hydrocarbons and then that data will be of any use. Additionally, DFT computations also have their issues, for example, calculations using gamma point cannot be trusted.

3) Authors show in supplementary Fig. 4 that biomass first breaks down due to hot water hydrolysis and then the catalytic process kicks in to break down the individual biomass components. However, in the experiments without the catalyst, you can see the solid ppt. This makes one believe that there is a role that the catalyst couple possibly be playing in breaking the biomass too. However, there is no investigation into this aspect.

Reviewer #2 (Remarks to the Author):

The manuscript reports remarkable results for the conversion of different biomass feedstock into bio natural gas using a Ni-Al based catalyst under hydrogen atmosphere and autogenous pressure. These

conditions affords a very high conversion of the biomass as well as a high methane yield. Moreover, the catalysts keeps its activity even after 30 reaction cycles.

Accordingly, these results can be relevant in terms of both scientific and technological impacts.

However, there are a number of important issues that need to be addressed:

- It is unclear which is the pressure range reached during the operation of the reaction. The authors indicate that the reactor is cold loaded with 4 MPa of H₂. However, after heating at temperatures up to 300 °C, the pressure should be significantly higher. This data is important to assess the feasibility of this process according to safety and economic aspects.
- The H₂ consumption during the reaction should be experimentally determined from the composition of the gaseous phase, and not just estimated from mass balances, at least for some catalytic tests.
- While the authors investigate the effect of the feedstock nature and reaction temperature, the influence of the hydrogen pressure is not studied. Again, this is a very relevant variable in this process, hence its influence should be checked.
- The authors state in the abstract and conclusions sections that 100% conversion of the raw biomass to bio-natural gas is reached. This maybe not completely right since, depending of the feedstock, both solid and liquid products are also obtained.
- The process scheme and method used for the techno-economic analysis and life cycle assessment are poorly described. It is not clear how the H₂ is produced, which is a very important issue. According to supplementary Fig. 9 it seems that hydrogen comes from fossil natural gas. In such case, it would not have much sense to transform fossil natural gas into hydrogen and then to use this hydrogen for bio-natural gas production.
- Moreover, in Figure 4 the authors propose that the produced bio-methane could be employed back for producing hydrogen. The energy and economic penalties of these cyclic transformations (methane -> hydrogen -> methane -> hydrogen) would be great.
- Finally, according to the own authors results , the environmental benefits of this technology would be modest, with just reductions of 34% and 26% of primary energy fossil energy depletion and greenhouse gas emissions compared to fossil-natural gas. At present, there are other technologies rather more efficient for the transformation of biomass feedstocks into fuels.

In summary, although the results are scientifically interesting, important concerns appear on the feasibility of the investigated route for the conversion of biomass into bio-methane. In this way, it would be convenient that the authors compares as well the performance of this process with the traditional route based on anaerobic digestion followed by separation/purification of the so-produced methane.

Reviewer 1:

Comment 1:

Any form of lignocellulosic Biomass contains about 30-40 wt% of oxygen and when you try to transform that biomass into any form, that oxygen is going to be present in one of the products/by-products (to complete oxygen balance). However, I was intrigued to see the data in Fig. 1, where after the process, at 300, authors claim that they have about 90% yield of gas that contains mostly C1-C4 hydrocarbons and little amount of CO₂. I fail to understand where does the large amount of oxygen that is present in the biomass go??? Based on the yield of solid products and some characterization they have shown, its not possible to have all that oxygen in the solid. This raises serious concern about their analysis methods. Authors must demonstrate clearly where is the oxygen gone.

Response 1:

Thanks very much for your detailed comments. In the catalytic conversion process, most oxygen in lignocellulosic biomass was combined with hydrogen to be converted to H₂O, which was also the solvent in the reaction.

In the catalytic conversion of beech sawdust, small amount of oxygen was converted to CO₂ in gas products and oxygenated organic compounds in liquid products, respectively. Most oxygen in beech sawdust was converted to H₂O under hydrogen atmosphere, which consumed the hydrogen from lignocellulosic biomass itself and extra H₂ filled in the autoclave.

After the reaction at 300 °C for 5 h, the carbon yield of liquid products was only 0.1% which almost could be neglected. However, the total weight of the liquid fraction after reaction increased by 0.34 g, which further demonstrated the weight of solvent H₂O increased. The extra H₂O came from the oxygen in lignocellulosic biomass combined with hydrogen.

Ultimately, according to the amount of CO₂ in gas products, 24% of the oxygen in beech sawdust was converted to CO₂, and the oxygenated compounds in liquid products with the carbon yield of only 0.1% almost could be neglected. Thus, nearly

76% of the oxygen in beech sawdust was converted to H₂O, which was basically consistent with the increased weight of solvent H₂O.

Comment 2:

Though the process is interesting and intriguing, there is little science presented in the MS. For example, the DFT calculations are absolutely of no use. In the experiment, they show that without catalyst, the sawdust remains in the liquid solution and with catalyst, you see it getting converted. And then they show DFT data showing some elementary steps of ethanol dissociation (without any kinetics though) and claim that those steps are favored thermodynamically. However, catalyst's job is not to alter the thermodynamics but to alter the kinetics. Yes, some steps could be downhill in energy on one catalysts vs the other, but overall thermodynamics cannot be altered by the catalyst. Authors need to clearly show how kinetics of the steps involved in biomass molecules transformation into smaller hydrocarbons and then that data will be of any use. Additionally, DFT computations also have their issues, for example, calculations using gamma point cannot be trusted.

Response 2:

Thank you for your comments. Indeed, our process of catalytic conversion of lignocellulosic biomass to natural gas is interesting and intriguing. The agricultural and forestry residues mainly involved cellulose, hemicellulose and lignin, which were extremely complicated natural polymers. Thus, we try to use simple model compounds to peek a little bit of information about the actual reaction process. The experiment results showed that the gas carbon yield for the conversion of methanol was much higher than that of ethanol (Supplementary Table 3), which demonstrated that C-O bond was more easily to break than C-C bond in our reaction system. And the higher carbon yield of natural gas was acquired over the nickel-based catalyst with Ni₂Al₃ alloy phase than that of commercial nickel catalyst for both methanol and ethanol molecules. Thus, nickel-based catalyst with Ni₂Al₃ phase exhibited the high catalytic activity for the cleavage of C-O and C-C bonds.

And then, we tried to use DFT theoretical calculations to provide some corroborative

evidence about the adsorption of model compound on the catalyst surface. We used a simplified calculation to analyze the most likely intermediate active species in the key steps. By analyzing the adsorption state energy at key points in the entire potential energy surface of Ni(111) and Ni₂Al₃(110) surfaces to provide hints to help us speculate the reaction mechanism on cleavage of C-O and C-C bonds. According to the energy change in these pathways, it showed that the preferentially adsorption of hydroxyl groups on aluminum sites of Ni₂Al₃(110) made the C-O cleavage occurring seems more readily on Ni₂Al₃(110) than that on Ni(111), which might further promoted the cleavage of C-C bond. The above analysis was similar with the experimental results. And these simple DFT calculations were also used in the reported literatures to demonstrate their results (Min, W. et al. *Nat. Commun.* **2020**, 11, 1083; Zhe, Z. et al. *J. Am. Chem. Soc.* **2021**, 143, 6533). We made corresponding changes to the problem of calculations using gamma point (Fig. R1, R2 and R3).

Fig. R1 Reaction pathways of methanol compound. **a, b**, Energy profiles for CH₃OH dissociation on Ni(111) and Ni₂Al₃(110) surfaces. The x axis shows the reaction intermediates, the y axis shows the relative energy of each state.

Fig. R2 Reaction pathways of ethanol compound. a, b, Energy profiles for $\text{CH}_3\text{CH}_2\text{OH}$ dissociation on Ni(111) and $\text{Ni}_2\text{Al}_3(110)$ surfaces of C-C bonds preferentially cleaved. The x axis shows the reaction intermediates, the y axis shows the relative energy of each state.

Fig. R3 Reaction pathways of ethanol compound. Energy profiles for $\text{CH}_3\text{CH}_2\text{OH}$ dissociation on Ni(111) and $\text{Ni}_2\text{Al}_3(110)$ surfaces of C-O bonds preferentially cleaved. The x axis shows the reaction intermediates, the y axis shows the relative energy of each state.

Additionally, we further performed the kinetic analysis of methanol and ethanol conversion over nickel based catalyst (Fig. R4). The apparent activation energies (E_a) of methanol and ethanol transformation upon using nickel-based alloy catalyst were 114.0 and 121.7 kJ mol^{-1} , respectively, which was lower than using commercial nickel catalyst, manifesting the relative catalytic role of Ni_2Al_3 alloy in promoting the cleavage of C-O and C-C bonds.

Fig. R4 Kinetic measurements of methanol and ethanol conversion. Arrhenius

plots for **a**, methanol, **b**, ethanol over nickel based catalysts.

Comment 3:

Authors show in supplementary Fig. 4 that biomass first breaks down due to hot water hydrolysis and then the catalytic process kicks in to break down the individual biomass components. However, in the experiments without the catalyst, you can see the solid ppt. This makes one believe that there is a role that the catalyst couple possibly be playing in breaking the biomass too. However, there is no investigation into this aspect.

Response 3:

We totally agree with your suggestion. In the catalytic conversion of lignocellulosic biomass, beech sawdust existed in the form of solid in the solvent of H₂O without the catalyst, the addition of nickel-based alloy catalyst showed nearly complete conversion of the solid under the same reaction condition. Although the literature reports that hot water can hydrolyze biomass, it cannot be ruled out that water and catalyst work together to break the raw biomass. We made corresponding changes to the manuscript.

Reviewer 2:

Comments 1:

It is unclear which is the pressure range reached during the operation of the reaction. The authors indicate that the reactor is cold loaded with 4 MPa of H₂. However, after heating at temperatures up to 300 °C, the pressure should be significantly higher. This data is important to assess the feasibility of this process according to safety and economic aspects.

Response 1:

Thanks very much for your comments. In the catalytic conversion of lignocellulosic biomass, the reaction pressure during the reaction was important to assess this process and shown in Fig. R5, which ranged from 10.3 to 12.4 MPa at 300 °C.

Additionally, in our paper, the techno-economic analysis was conducted on the basis of the maximum pressure value of 12.4 MPa and the final pressure value of 10.3 MPa at 300 °C. Thus, the reaction pressure during the reaction had been considered for the safety and economic aspects in our paper.

Fig. R5 The reaction pressure during the reaction. Reaction condition: 1.0 g beech sawdust, 5.2 mmol nickel-based alloy catalyst, 20 mL H₂O, 4 MPa H₂, 300 °C, 5 h.

Comment 2:

The H₂ consumption during the reaction should be experimentally determined from

the composition of the gaseous phase, and not just estimated from mass balances, at least for some catalytic tests.

Response 2:

Thank you for the comments. According to the experiment results, the molar ratio of consumed H₂ to generated carbon in gaseous phase was 1.6 and shown in Supplementary Table 9.

In the catalytic conversion of lignocellulosic biomass, before the reaction, the autoclave was filled with 4.0 MPa H₂ at room temperature. The initial amount of H₂ could be calculated by the Ideal Gas Equation. Then the reaction was conducted at 300 °C for 5 h with beech sawdust as the substrate, and the composition of the gaseous phase after the reaction was quantified by using tandem gas chromatography (GC) with thermal conductivity detector (TCD) and flame-ionization detector (FID). Thus, the H₂ consumption during the reaction could be calculated according the amounts of H₂ before and after the reaction, which was 5.1×10^{-2} mol. And the amount of generated carbon in gaseous phase after the reaction was quantified to be 3.1×10^{-2} mol. Finally, the molar ratio of consumed H₂ to generated carbon in gaseous phase was 1.6, which was calculated based on the experiment results.

Comment 3:

While the authors investigate the effect of the feedstock nature and reaction temperature, the influence of the hydrogen pressure is not studied. Again, this is a very relevant variable in this process, hence its influence should be checked.

Response 3:

We further investigated the effect of the hydrogen pressure on the gas products, and the detailed gas distribution was depicted in Fig. R6. As the initial H₂ pressure increased from 0 to 4 MPa, the carbon yield of gas products increased from 74.0% to 87.2%. Thereinto, the distribution of CH₄ increased from 51.4 to 86.1 mol%, and the distribution of CO₂ decreased from 45.8 to 9.9 mol%. Thus, the content of CH₄ in gas products increased with the increasing initial H₂ pressure.

Fig. R6 Carbon yield of gas products with the different H₂ pressure. Reaction condition: 1.0 g beech sawdust, 5.2 mmol nickel-based alloy catalyst, 20 mL H₂O, 300 °C, 5 h.

Comment 4:

The authors state in the abstract and conclusions sections that 100% conversion of the raw biomass to bio-natural gas is reached. This maybe not completely right since, depending of the feedstock, both solid and liquid products are also obtained.

Response 4:

We are sorry for the inaccurate description of the conversion process of raw biomass to bio-natural gas. In the catalytic conversion of lignocellulosic biomass, the gas, liquid and solid products were obtained simultaneously. Overall, the gas products were the main reaction products with the highest carbon yield. The sentence of inaccurate description was modified in our article.

Comment 5:

The process scheme and method used for the techno-economic analysis and life cycle assessment are poorly described. It is not clear how the H₂ is produced, which is a very important issue. According to supplementary Fig. 9 it seems that hydrogen comes from fossil natural gas. In such case, it would not have much sense to

transform fossil natural gas into hydrogen and then to use this hydrogen for bio-natural gas production.

Response 5:

Thank you for your comments. The process scheme and method used for the techno-economic analysis and life cycle assessment are further described in detail. The bio-natural gas production requires hydrogen, and hydrogen comes from fossil natural gas. It seems to have little sense to transform fossil natural gas into bio-natural gas production. But from the view of production inventory, 1 kg of bio-natural gas requires 0.23 kg of hydrogen, while 1 kg hydrogen production requires 3.36 kg of natural gas. It implies that 1 kg of bio-natural gas requires 0.77 kg of fossil natural gas, and 0.23 kg of bio-natural gas was created with the hydrogen from lignocellulosic biomass itself.

In addition, hydrogen production is a hotspot and many sources has been reported, such as coal gasification, natural gas reforming, biomass gasification, biomass reforming, biomass electrolysis, nuclear thermochemical, nuclear electrolysis, solar electrolysis, and wind electrolysis. Integrating hydrogen production with non-fossil pathways can dramatically reduce primary fossil energy depletion and greenhouse gases emissions. However, the natural gas reforming is selected in view of its widespread application at present, which can identify opportunities and constraints associating with the current work rather than advanced hydrogen production pathway. The results indicates that bio-natural gas derived from natural gas-based hydrogen is prior compared to natural gas, which implies that bio-natural gas derived from non-fossil resource-based hydrogen will generate more potential with saving energy and GHG emission reduction.

Comment 6:

Moreover, in Figure 4 the authors propose that the produced bio-methane could be employed back for producing hydrogen. The energy and economic penalties of these cyclic transformations (methane -> hydrogen -> methane -> hydrogen) would be great.

Response 6:

We agree with this point. Although, the hydrogen transportation now is primarily depended on high pressure tube trailers and liquid hydrogen tank which leads to high cost and a certain risks, storing hydrogen in natural gas could greatly decrease transportation cost of hydrogen, and enabled the gas supply to the end user by the large-scale transportation pipelines and storage stations of natural gas around the world. But, The energy and economic penalties of these cyclic transformations (methane -> hydrogen -> methane -> hydrogen) would be existing. We made corresponding changes to the manuscript.

Comment 7:

Finally, according to the own authors results, the environmental benefits of this technology would be modest, with just reductions of 34% and 26% of primary energy fossil energy depletion and greenhouse gas emissions compared to fossil-natural gas. At present, there are other technologies rather more efficient for the transformation of biomass feedstocks into fuels.

Response 7:

Undeniably, there are several other efficient technologies for the transformation of biomass feedstocks into fuels currently. But emerging technologies might generate considerable potential benefit with development of society economy. For example, hydrogen can produced from many sources, such as coal gasification, natural gas reforming, biomass gasification, biomass electrolysis, nuclear thermochemical, nuclear electrolysis, solar electrolysis, and wind electrolysis. Different hydrogen sources will generate varied results. Fig. R7 is used to depict the effect of life cycle primary fossil energy depletion and greenhouse gas emissions on different hydrogen sources. It can be seen that integrating hydrogen production with non-fossil pathways can dramatically reduce primary fossil energy depletion and greenhouse gases emissions.

Fig. R7 Effects of hydrogen source on life cycle GHG emissions and PFE depletion of bio-natural gas.

REVIEWER COMMENTS

Reviewer #1 (Remarks to the Author):

Except, the first comment that I had made about the oxygen content, authors have NOT addressed any of my comments to my satisfaction. They have tried explaining the same thing that's there in the MS to justify that what they have done is sufficient. However, I cannot recommend publication of this paper in journal like Nature Comm, unless authors perform detailed kinetic study with all the transition states and free energy barriers for the entire reaction pathway. Authors justify that such minimal calculations are good enough for publishing their paper in Nature Comm by giving some previous examples. But every paper is different and in this paper I cannot recommend it for publication in this form.

I had raised an issue about gamma point calculation and the authors' response was "We made corresponding changes to the problem of calculations using gamma point". What does that mean? Did you perform bench-marking calculations with multiple K-points? How many K-points did you use and why???

No investigation into the role of catalyst in breaking down the biomass (my comment3). Authors agreed to it but never investigated it.

Reviewer #2 (Remarks to the Author):

The authors have partially addressed the comments of this reviewer for the preparation of the revised version. In particular, the study performed varying the initial hydrogen pressure is interesting.

However, a relevant issue is still unsolved. The authors still consider methane steam reforming as the route for the production of hydrogen. As a consequence, and according to the theoretical estimation of the own authors, 0.77 kg of fossil natural gas will be needed to produce 1 kg of bio-methane. Moreover, this transformation will require to carry out a number of operations: steam reforming of natural gas, separation and purification of hydrogen, hydrothermal treatment of the lignocellulosic biomass with hydrogen under relatively high pressures, separation and purification of the produced bio-methane, etc. It is evident that this scheme is little feasible to be accomplished at large scale with both environmental and economic benefits. Thus, the own authors conclude that this strategy would lead at best to just 26% reduction of greenhouse gas emissions compared to the direct use of fossil natural gas.

My recommendation is to consider in the analyses other sources of hydrogen, mainly renewable sources as water electrolysis with electricity from wind or solar. Perhaps in this case, it would be possible to get a rather better reduction of CO₂ emissions. Likewise, the authors could change the reactor pressure in order to reduce it somewhat as it may also decrease the energy input of the process. On the contrary, with the current data the proposed process does not have much sense.

Other comments:

- The new data incorporated to the revised version indicate that 1.6 mol of H₂ are consumed per mol of C in the gases, while the theoretical value for producing CH₄ should be 2. On the other hand, a great part of H₂ should be also consumed in the removal of oxygen from lignocellulose and thus producing water. Therefore, I think that the overall hydrogen consumption should be quite larger than the estimated one.
- What does it mean "distribution of CH₄" and "distribution of CO₂" in response 3? Concentration?
- It would be convenient to include Fig.R5 in the supporting information as it provides the evolution along the time of the actual reactor pressure.

Reviewer 1:

Comment 1:

I had raised an issue about gamma point calculation and the authors' response was "We made corresponding changes to the problem of calculations using gamma point". What does that mean? Did you perform bench-marking calculations with multiple K-points? How many K-points did you use and why???

Response 1:

Thanks very much for your comments. We further perform the detailed kinetic study with all the transition states and free energy barriers for the entire reaction pathways. In this work, we tested the convergence of k-point grid by calculating the reaction energy of $\text{CH}_3\text{CH}_2\text{OH}^* + * \rightarrow \text{CH}_3\text{CH}_2\text{O}^* + \text{H}^*$ on the $\text{Ni}_2\text{Al}_3(110)$ surface with various k-point grids - 1x1x1, 1x2x1, 2x3x1, 3x4x1 and 4x5x1. The reaction energy converges at the 2x3x1 k-point grid. The 2x3x1 k-point grid was chosen for the calculations. The results showed that the decomposition of $\text{CH}_3\text{CH}_2\text{OH}^*$ into $\text{CH}_3\text{CH}_2\text{O}^*$ was thermodynamically favored on Ni(111) and $\text{Ni}_2\text{Al}_3(110)$, with energy barriers of 1.04 eV and 0.53 eV, respectively. For the further cleavage of C-O and C-C bonds, two main reaction pathways were evaluated, which were C-C bond preferentially cleaved ($\text{CH}_3\text{CH}_2\text{O}^* \rightarrow \text{CH}_3^* + \text{CH}_2\text{O}^*$) and C-O bond preferentially cleaved ($\text{CH}_3\text{CH}_2\text{O}^* \rightarrow \text{CH}_3\text{CH}_2^* + \text{O}^*$). However, the process of C-C bond preferentially cleaved ($\text{CH}_3\text{CH}_2\text{O}^* \rightarrow \text{CH}_3^* + \text{CH}_2\text{O}^*$) was thermodynamically hindered on Ni(111) and $\text{Ni}_2\text{Al}_3(110)$ (Fig. R2), and the energy barriers reached up to 2.14 eV and 2.41 eV, respectively. In contrast, the process of C-O bond preferentially cleaved ($\text{CH}_3\text{CH}_2\text{O}^* \rightarrow \text{CH}_3\text{CH}_2^* + \text{O}^*$) was more feasible (Fig. R1). Furthermore, the energy barrier for $\text{CH}_3\text{CH}_2^* \rightarrow \text{CH}_3^* + \text{CH}_2^*$ process on $\text{Ni}_2\text{Al}_3(110)$ was 1.60 eV, which was much lower than that of 2.09 eV on Ni(111). Thus, the above results demonstrated that the preferential cleavage of C-O bond in $\text{CH}_3\text{CH}_2\text{O}^*$ on aluminum sites of $\text{Ni}_2\text{Al}_3(110)$ might promote the further cleavage of C-C bond in CH_3CH_2^* intermediate.

Fig. R1 Energy profiles for CH₃CH₂OH dissociation on Ni(111) and Ni₂Al₃(110) surfaces of C-O bond preferentially cleaved. The *x* axis shows the reaction intermediates, the *y* axis shows the relative energy of each state.

Fig. R2 Energy profiles for CH₃CH₂OH dissociation on Ni(111) and Ni₂Al₃(110) surfaces of C-C bond preferentially cleaved. The *x* axis shows the reaction intermediates, the *y* axis shows the relative energy of each state.

Fig. R3 The side and top view of reaction species on Ni(111) and Ni₂Al₃(110).

Comment 2:

No investigation into the role of catalyst in breaking down the biomass (my comment 3). Authors agreed to it but never investigated it.

Response 2:

Thanks very much for your comments. The effect of H₂O and nickel-based alloy catalyst on the catalytic conversion of raw biomass to natural gas was investigated, and the detailed reaction condition and results were listed in Supplementary Table 2. As shown in Fig. R4, when the experiment was conducted without catalyst in solvent H₂O, it mainly generated CO₂ with 5.3% of gas carbon yield. Whereas, when the solvent H₂O was replaced by the nonpolar n-tetradecane, the experiment was conducted without H₂O and obtained the gas products of CH₄, CO₂ and C₂-C₄ hydrocarbons with 20.7% of gas carbon yield. Ultimately, 49.4% of gas carbon yield was acquired from the catalytic conversion of beech sawdust with nickel-based alloy catalyst in solvent H₂O under the same operating conditions. Thus, H₂O and nickel-based alloy catalyst acted in synergy to break the biomass and generate the higher carbon yield of gas products.

Fig. R4 Effects of H₂O and catalyst on the catalytic conversion of raw biomass to bio-natural gas.

Reviewer 2:

Comment 1:

However, a relevant issue is still unsolved. The authors still consider methane steam reforming as the route for the production of hydrogen. As a consequence, and according to the theoretical estimation of the own authors, 0.77 kg of fossil natural gas will be needed to produce 1 kg of bio-methane. Moreover, this transformation will require to carry out a number of operations: steam reforming of natural gas, separation and purification of hydrogen, hydrothermal treatment of the lignocellulosic biomass with hydrogen under relatively high pressures, separation and purification of the produced bio-methane, etc. It is evident that this scheme is little feasible to be accomplished at large scale with both environmental and economic benefits. Thus, the own authors conclude that this strategy would lead at best to just 26% reduction of greenhouse gas emissions compared to the direct use of fossil natural gas.

My recommendation is to consider in the analyses other sources of hydrogen, mainly renewable sources as water electrolysis with electricity from wind or solar. Perhaps in this case, it would be possible to get a rather better reduction of CO₂ emissions. Likewise, the authors could change the reactor pressure in order to reduce it somewhat as it may also decrease the energy input of the process. On the contrary, with the current data the proposed process does not have much sense.

Response 1:

Thanks very much for your detailed comments. We further investigate the effect of hydrogen source, such as coal gasification, natural gas reforming, solar electrolysis, nuclear electrolysis and biomass gasification, on the life cycle primary fossil energy (PFE) depletion and greenhouse gas (GHG) emissions of bio-natural gas. As shown in Fig. R5, it revealed that the integrated hydrogen production with non-fossil pathways could dramatically reduce PFE depletion and GHG emissions. Specifically, the production of bio-natural gas using H₂, which was derived from solar electrolysis, nuclear electrolysis and biomass gasification, could reduce 76.5-88.1% of PFE depletion and 67.9-81.9% of GHG emissions compared to natural gas

refroming-based H₂. Moreover, the above pathways could reduce 81.4-90.5% of PFE depletion and 76.1-86.5% of GHG emissions compared to the conventional coal gasification-based H₂.

Fig. R5 Effects of hydrogen source on life cycle PFE depletion and GHG emissions of bio-natural gas.

Additionally, as hydrogen accounts for the predominant contribution to the production of bio-natural gas, the effect of hydrogen pressure on techno-economic analysis (TEA) and life cycle assessment (LCA) of bio-natural gas was also investigated. As shown in Fig. R6, when the hydrogen pressure in the catalytic reaction system decreased from 4 to 0.1 MPa, the production cost of natural gas decreased from 4100 to 2625 CNY t⁻¹, meanwhile, the PFE depletion and GHG emissions generally reduced to only 6.2% and 9.5% of that with 4 MPa hydrogen for bio-natural gas. Thus, the hydrogen pressure played a vital role in the TEA and LCA of bio-natural gas, and the decreased hydrogen pressure would boost the bio-natural gas production for a low-carbon footprint.

Fig. R6 Effects of hydrogen pressure on TEA and LCA of bio-natural gas.

Comments 2:

The new data incorporated to the revised version indicate that 1.6 mol of H₂ are consumed per mol of C in the gases, while the theoretical value for producing CH₄ should be 2. On the other hand, a great part of H₂ should be also consumed in the removal of oxygen from lignocellulose and thus producing water. Therefore, I think that the overall hydrogen consumption should be quite larger than the estimated one.

Response 2:

Thanks very much for your comments. As we all know, lignocellulosic biomass contains a large amount of H except for the C and O. In our work, the weight percentage of C, H and O in beech sawdust was 43.0%, 7.2% and 39.5%, respectively, and the mole ratio of C:H:O was 3:6:2. In the catalytic conversion of beech sawdust, a part of H in beech sawdust was consumed in the removal of oxygen from lignocellulose. Meanwhile, there also existed the left H in beech sawdust, which could be utilized to combine with C to generate CH₄. Thus, the mole ratio of consumed H₂ to generated carbon in gas phase was less than 2, which was in agreement with our experimental value of 1.6.

Comments 3:

What does it mean "distribution of CH₄" and "distribution of CO₂" in response 3?

Concentration?

Response 3:

We are sorry for the inaccurate description. The distribution of CH₄ or CO₂ was modified as the molar percentage of CH₄ or CO₂ in gas products. And the gas products distribution was calculated according to the mole numbers of each gas and the total mole numbers of gas products, which was defined in our manuscript as follows:

$$\text{Gas products distribution} = \left(\frac{\text{mole numbers of each gas}}{\text{total mole numbers of gas products}} \right) \times 100\%$$

Comments 4:

It would be convenient to include Fig.R5 in the supporting information as it provides the evolution along the time of the actual reactor pressure.

Response 4:

Thanks very much for your detailed comments. The revised supporting information included Fig. R5 about the actual reactor pressure during the reaction, which was listed as Supplementary Fig. 11.

REVIEWERS' COMMENTS

Reviewer #1 (Remarks to the Author):

Authors have addressed my concerns and I am happy to recommend the paper for publication now.

Reviewer #2 (Remarks to the Author):

The authors have conveniently addressed the comments of the reviewers.

The new data included in the work have confirmed the great effect of both the operating pressure and the H₂ source, as anticipated in the review, improving sharply the overall feasibility and environmental impacts of the process in comparison with the previous section.

Accordingly, I think that the corresponding figures (R5 and R6) should be included directly in the article rather than just in the supporting information, while those effects should be highlighted in the abstract. On the other hand, the extension of the Discussion section looks too short (just have a page). Then, I propose to merge Results and Discussion sections.

Reviewer 1:

Authors have addressed my concerns and I am happy to recommend the paper for publication now.

Response:

We appreciate your valuable remarks and suggestions, which indeed helped us to significantly improve the quality of this work.

Reviewer 2:

The authors have conveniently addressed the comments of the reviewers.

The new data included in the work have confirmed the great effect of both the operating pressure and the H₂ source, as anticipated in the review, improving sharply the overall feasibility and environmental impacts of the process in comparison with the previous section.

Accordingly, I think that the corresponding figures (R5 and R6) should be included directly in the article rather than just in the supporting information, while those effects should be highlighted in the abstract.

On the other hand, the extension of the Discussion section looks too short (just have a page). Then, I propose to merge Results and Discussion sections.

Response:

Thanks very much for your detailed comments. The revised article included Fig. R5 and R6, which were listed as Fig. 3d and 3e. And the effect of hydrogen operating pressure and the hydrogen source was further highlighted in the Abstract.

Moreover, according to the reviewer's suggestions, Results and Discussion sections were merged in the revised article.